# Stacked Filters Stationary Flow For Hardware-Oriented Acceleration Of Deep Convolutional Neural Networks

**Gao Yuechao, Liu Nianhong & Zhang Sheng** [*]
Department of Microelectrics and Nanoelectrics
Tsinghua University
Beijing, 100084, China
`{gyc15,lnh15}@mails.tsinghua.edu.cn`

## Abstract

To address memory and computation resource limitations for hardware-oriented acceleration of deep convolutional neural networks (CNNs), we present a computation flow, stacked filters stationary flow (SFS), and a corresponding data encoding format, relative indexed compressed sparse filter format (CSF), to make the best of data sparsity, and simplify data handling at execution time. Comparing with the state-of-the-art result (Han et al., 2016b), our methods achieve $1.11\times$ improvement in reducing the storage required by AlexNet, and $1.09\times$ improvement in reducing the storage required by SqueezeNet. Moreover, using these approaches, chip area for logics handling irregular sparse data access can be saved. Comparing with the 2D-SIMD processor structures in DVAS, ENVISION, the processing element (PE) array utilization rate improves from 26.4% to 96.5%, using our methods on the data from Deep Compression on AlexNet.

## 1 Introduction

CNNs have achieved substantial progress during the past years. But hardware resource limitations have hindered their wide usage in embedded devices. Various efforts have been made to address this issue, such as ShiftCNN (Gudovskiy & Rigazio, 2017), Ristretto (Gysel, 2016), Eyeriss (Chen et al., 2017), Deep Compression (Han et al., 2016b) and EIE (Han et al., 2016a). Through compressing deep neural networks with pruning, trained quantization and Huffman coding, Deep Compression (Han et al., 2016b), the best paper of ICLR 2016, achieved state-of-the-art result in reducing storage requirement of neural networks without affecting their accuracy.

In spite of the great progress achieved till now, there are still many problems to be solved. The first problem is manipulating compressed sparse data need considerable extra logics. Eyeriss uses network on chip to handle sparsity by only performing data reads and MACs on nonzero values; DVAS (Moons & Verhelst, 2015) and ENVISION (Moons et al., 2017) use input guard memories and guard control units to handle data sparsity. Several sparse matrix encoding formats have been proposed, such as CSC, CSR and CISR (Fowers et al., 2014). But existing encoding formats complex the computation at runtime due to their irregular memory access characteristics. This results in inefficiency in parallelizing computation and bigger chip area. For example, EIE use Pointer Read Units (accounting for about 19.1% chip area) and a Sparse Matrix Read Unit (accounting for about 73.57% chip area) to handle compressed sparse data. Therefore, it would be desirable if the sparse data can be easily handled during execution without complex transformations, lookups and computation. The second one is, for deeply compressed sparse networks, the PE array utilization rate of recently proposed hardware acceleration designs, such as Eyeriss, DVAS, ENVISION, DNPU (Shin et al., 2017), etc., is fairly low. In this paper we present a novel computation flow SFS, and a corresponding data encoding format CSF that data can be straightforwardly handled at run time. We also propose a three dimensional Single Instruction Multiple Data (3D-SIMD) processor architecture to illustrate how to accelerate deep CNNs by taking advantage of the SFS flow and CSF format.

---

[*]Corresponding author. zhang_sh@tsinghua.edu.cn

## 2 STACKED FILTERS STATIONARY FLOW (SFS) AND RELATIVE INDEXED COMPRESSED SPARSE FILTER (CSF) FORMAT

Computations of convolutional and fully connected layers in CNNs can be unified into one formula Eq.1 (ignoring biases). Eq. 2-6 illustrate the approach SFS. $V_o$, $V_i$ and $W_f$ are the matrices of output feature maps, input feature maps and filters, respectively. $S, C, K, M, M', m$ is a given stride size, channel number, filter kernel size, total filter number, number of batches and batch size. Filters are firstly grouped into $M'$ batches with batch size $m$, and each $W_f^{(n)}$ is then reshaped to $W_{f'}^{(n)}$. One channel of feature data will convolute with $m$ filters from the same channel in parallel (Eq.4, $j = 0, ..., m - 1$, pseudo code is illustrated in figure 2). At the end of computation, $V_{o'}^{(0)}$ - $V_{o'}^{(M'-1)}$ are concatenated back to $V_o$.

$$V_o[cho][y][x] = \sum_{chi=0}^{C-1} \sum_{r=0}^{K-1} \sum_{c=0}^{K-1} W_f[cho][chi][r][c] \times V_i[chi][Sy + r][Sx + c] \tag{1}$$

$$W_f = [W_f^{(0)}, W_f^{(1)}, ..., W_f^{(M'-1)}], W_{f'} = [W_{f'}^{(0)}, W_{f'}^{(1)}, ..., W_{f'}^{(M'-1)}] \tag{2}$$

$$W_{f'}^{(n)}[chi][r][c][j] = W_f^{(n)}[j][chi][r][c] \tag{3}$$

$$V_{o'}^{(n)}[j][y][x] = \sum_{chi=0}^{C-1} \sum_{r=0}^{K-1} \sum_{c=0}^{K-1} W_{f'}^{(n)}[chi][r][c][j] \times V_i[chi][Sy + r][Sx + c] \tag{4}$$

$$V_o = [V_{o'}^{(0)}, V_{o'}^{(1)}, ..., V_{o'}^{(M'-1)}] \tag{5}$$

$$M' = M/m, 0 \le n < M', 0 \le j < m, 0 \le cho < M. \tag{6}$$

As to the encoding format CSF, this approach is to further rearrange the memory layout of the grouped $m$ filters illustrated in figure 1-a, storing the elements column by column. So in computation flow SFS, when each element in the feature map multiplies with a column of data from $m$ filters (figure 2), the filter weights could be loaded sequentially. The first line in figure 1-b illustrates the changing. In figure 1-b, if there is any weight value equals to 0, just remove that value and its index, add one to the relative index of the next value, and subtract one to the pointer of the next column. The nonzero value number (includes padding zeros) of a column is given by the pointer of the next column. Column pointer is 0 means all the values in the column before the column of this pointer equal to 0. Relative column pointer is not needed when parameters are stored in files.

| Filter 1 | W1,11 | W1,12 | W1,13 | W1,21 | ... | W1,33 | Virtual weight value | W1,11 | W2,11 | ... | Wm,11 | ... | W1,kk | W2,kk | ... | Wm,kk |
|---|---|---|---|---|---|---|---|---|---|---|---|---|---|---|---|---|
| ... | ... | ... | ... | ... | ... | ... | Relative filter index | 0 | 0 | ... | 0 | ... | 0 | 0 | ... | 0 |
| Filter m | Wm,11 | Wm,12 | Wm,13 | Wm,21 | ... | Wm,33 | Relative column pointer | 0 | m | ... | m | m | | | | |
| | | | a) | | | | | | | b) | | | | | | |

Figure 1: Memory layout a) for $m$ filters with kernel size 3 from a single channel. b) in CSF format.

## 3 3D-SIMD PROCESSOR ARCHITECTURE

The SFS flow and the CSF encoding format are two key features of the proposed 3D-SIMD processor architecture, see figure 3. In this architecture, after feature data are loaded into the line buffer and

```
for chi in [0, C −1], r in [0, K −1], c in [0, K −1] {
    outch 0 :  V_o'^(n)[0][y][x]+ = W_f'^(n)[chi][r][c][0]×V_i[chi][Sy + r][Sx + c]
    outch 1 :  V_o'^(n)[1][y][x]+ = W_f'^(n)[chi][r][c][1]×V_i[chi][Sy + r][Sx + c]
                          ...
    outch m −1: V_o'^(n)[m−1][y][x]+ = W_f'^(n)[chi][r][c][m−1]×V_i[chi][Sy + r][Sx + c]
}
```

Figure 2: SFS parallel computing pseudo code.    Figure 3: 3D-SIMD processor architecture.

window registers from a single channel of input feature map, and $m$ filter data from the same channel are loaded into the local filter buffer, each element in the window will multiply with a column of data from $m$ filters at the same position (figure 2). So data in CSF format can be straightforwardly handled without complex transformations, lookups and computation and loaded sequentially at run time, and zeros are skipped as designed. This demonstration shows that these two approaches can greatly simplify sparse data handling, saving zero bypassing and data lookup time. There are no complex sparse data handling logics needed comparing with former works (Moons & Verhelst, 2015; Moons et al., 2017; Han et al., 2016a).

## 4 RESULT

The distribution of continuous zero numbers after applying the changing is first evaluated. As figure 4 shows, the distribution narrows to the left. It means that fewer bits are needed to store the relative index values, and there will be fewer padding zeros when compressing data in encoding formats. This will further reduce storage space. The distribution of continuous nonzero numbers is also evaluated. It also narrows to the left, which means the computation load during execution will be better balanced comparing to the reference work (Han et al., 2016b). The effect of batch size $m$ on storage space is also analyzed. It shows that there do exist an optimum batch size for each layer. For simplicity, all the experiments in this section use filter number as the batch size. Table 1 illustrate the improvement of extra space needed to store index and padding zeros, and the improvement of total storage requirement after applying our method. The PE array utilization rate[1] improvement of convolutional and fully connected layers on several networks are also evaluated. On Alexnet, as illustrated in table 2, comparing with dense network processor like the 2D-SIMD processor structures in DVAS, ENVISION, etc., the PE array utilization rate improves from 26.4% to 96.5% (about $3.65\times$ improvement), using the data from Deep Compression on AlexNet[2]. The amount of data lookup calculation[3] is also evaluated. Using the same data above, the amount of calculation of SFS and CSF approach is about $1/20$ that of the algorithm in EIE (see table 2).

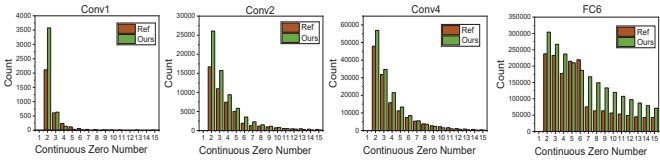

Figure 4: Distributions of continuous zero numbers on Alexnet, comparing with (Han et al., 2016b).

Table 1: Extra space(in bits) improvement and total storage requirement improvement

| Network | Nonzeros | Extra space | Improvement | Total |
|---|---|---|---|---|
| AlexNet by (Han et al., 2016b) | 30592056 | 41325956 | | |
| AlexNet by SFS+CSF | 30592056 | 34138970 | $1.21\times$ | $1.11\times$ |
| SqueezeNet by (Han et al., 2016b) | 3327368 | 1737628 | | |
| SqueezeNet by SFS+CSF | 3327368 | 1307160 | $1.33\times$ | $1.09\times$ |

Table 2: PE array utilization rate and data lookup calculation improvement (MACs in GOPS)

| | Total no. of MACs | No. of nonzero value MACs | Total no. of MACs (CSF) | Speed-up (CSF) | Lookup (CSF) |
|---|---|---|---|---|---|
| Alexnet CONV layers | 1.00269368 | 0.2744399 | 0.2839878 | $3.53\times$ | 1/13 |
| Alexnet FC layers | 0.05459595 | 0.0055178 | 0.0059305 | $9.21\times$ | 1/42 |
| Alexnet CONV+FC layers | 1.05728963 | 0.2799577 | 0.2899182 | $3.65\times$ | 1/20 |
| PE untilization ratio | 0.2647881 | | 0.9656438 | | |

---

[1]PE array utilization rate is estimated by: no. of nonzero value MACs / total no. of MACs.

[2]https://github.com/songhan/Deep-Compression-AlexNet

[3]Single calculation of locating a batch of data is defined as a basic unit.

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
