# OpenReview forum: "Stacked Filters Stationary Flow For Hardware-Oriented Acceleration Of Deep Convolutional Neural Networks"
_ICLR.cc/2018/Workshop — Accept_

### Official Review · AnonReviewer1 · 2018-03-09
**A method for reducing the space required for Hardware implementation of AlexNet**

**Rating:** 6
**Confidence:** 3

**Review:**

The authors present a new architecture for hardware implementation of AlexNet. The show a save in chip space (mainly the storage) and as a result in power.

The paper is written clearly. While the subject is important, the save factor they show is nice but not dramatic. The encoding format is important since besides having new architectures for deep learning, it is important to also to have some discussion on new sparse schemes.

In the overall, I recommend to accept this paper. Although the contribution is incremental, this paper is important for the hardware community in order to continue and develop methods for hardware implementations.

---

### Official Review · AnonReviewer2 · 2018-03-09
**Stacked filters for hardware acceleration**

**Rating:** 6
**Confidence:** 2

**Review:**

Stacked filters Stationary Flow For Hardware Oriented Acceleration of Deep Convolutional Neural Networks
Authors present a new stacked filter based architecture and a data encoding format for accelerating computations on CNNs.

Positives:
The stacked filters are an intuitive way to parallelize the computation
Paper emphasizes hardware efficiency:
PE usage is increased
Chip design can be simplified w/o the need for complex logic
Introduction seems to emphasize the pros and cons of competing technologies
Results show clear improvements
Negatives:
It is not apparent to me that their storage improvements are massive (though this is probably a very incremental field)
It is not clear to me whether or not this technique will help anybody using anything other than their special architecture
Notation in the math section is garbage
3D-SIMD section should be cut - does not follow well and the figure 3 is useless
Figure 1 is pretty non intuitive
The language is filled with run-ons and sentence fragments and generally unclear pronouns
The pseudocode is little more than a restatement of their horrendous math notation
Results are very hard to read
Figures are overall too small - need to focus or design better figures with this space limitation

---

### Official Review · AnonReviewer3 · 2018-03-10
**New ideas but lack real implementation**

**Rating:** 6
**Confidence:** 4

**Review:**

The three ideas in the paper are new. The proposed sparse storage format CSF makes data access easier than CSC, etc. The simulation results are promising comparing to the work by Han (2016). The result numbers are theoretic numbers. Please correct me if I'm wrong, but it looks to me that the authors have't implemented those ideas into a software package. It will be good know what the impact of these ideas is when using in real world applications.

---

> ### Public Comment · ~Yuechao_Gao1 · 2018-03-21
> **The impact of these ideas in real world applications**
>
> Thanks for your comments. Actually we are working on an FPGA implimentation. Comparing with the state-of-the-art results, our methods achieve 2× improvement for the computation efficiency per PE on most CONV and FC layers. Especially, our methods achieve 8× improvement for the computation efficiency per PE on Alexnet layer CONV4 with 384 filters, and 11× improvement on VGG16 layer CONV5-3 with 512 filters.

---

### Decision · Program_Chairs · 2018-03-20
**ICLR 2018 Workshop Acceptance Decision**

**Decision:**

Accept

**Comment:**

Congratulations, your paper was accepted to the ICLR workshop.